# The association between socioeconomic disadvantage and children's working memory abilities: A systematic review and meta-analysis

**Kate E. Mooney**[1]\*, **Stephanie L. Prady**[1], **Mary M. Barker**[2], **Kate E. Pickett**[1], **Amanda H. Waterman**[3,4]

**1** Department of Health Sciences, University of York, York, England, **2** Diabetes Research Centre, University of Leicester, Leicester, England, **3** School of Psychology, University of Leeds, Leeds, England, **4** Centre for Applied Education Research, Wolfson Centre for Applied Health Research, Bradford, England

\* kate.mooney@york.ac.uk

**Data Availability Statement:** All relevant data are within the manuscript and its supporting information files.

## Abstract

### Background and objective

Working memory is an essential cognitive skill for storing and processing limited amounts of information over short time periods. Researchers disagree about the extent to which socio-economic position affects children's working memory, yet no study has systematically synthesised the literature regarding this topic. The current review therefore aimed to investigate the relationship between socioeconomic position and working memory in children, regarding both the magnitude and the variability of the association.

### Methods

The review protocol was registered on PROSPERO and the PRISMA checklist was followed. Embase, Psycinfo and MEDLINE were comprehensively searched via Ovid from database inception until 3$^{rd}$ June 2021. Studies were screened by two reviewers at all stages. Studies were eligible if they included typically developing children aged 0–18 years old, with a quantitative association reported between any indicator of socioeconomic position and children's working memory task performance. Studies were synthesised using two data-synthesis methods: random effects meta-analyses and a Harvest plot.

### Key findings

The systematic review included 64 eligible studies with 37,737 individual children (aged 2 months to 18 years). Meta-analyses of 36 of these studies indicated that socioeconomic disadvantage was associated with significantly lower scores working memory measures; a finding that held across different working memory tasks, including those that predominantly tap into storage ($d$ = 0.45; 95% CI 0.27 to 0.62) as well as those that require processing of information ($d$ = 0.52; 0.31 to 0.72). A Harvest plot of 28 studies ineligible for meta-analyses further confirmed these findings. Finally, meta-regression analyses revealed that the association

**Funding:** KEP, AHW, and SLP's involvement was supported by the National Institute for Health Research Yorkshire and Humber ARC (reference: NIHR200166). The views expressed in this publication are those of the authors and not necessarily those of the National Institute for Health Research or the Department of Health and Social Care. The work of the lead author (KEM) was supported by an ESRC White Rose Doctoral Training Partnership Pathway Award. The funders had no role in study design, data collection and analysis, decision to publish, or preparation of the manuscript.

**Competing interests:** The authors have declared that no competing interests exist.

between socioeconomic position and working memory was not moderated by task modality, risk of bias, socioeconomic indicator, mean age in years, or the type of effect size.

## Conclusion

This is the first systematic review to investigate the association between socioeconomic position and working memory in children. Socioeconomic disadvantage was associated with lower working memory ability in children, and that this association was similar across different working memory tasks. Given the strong association between working memory, learning, and academic attainment, there is a clear need to share these findings with practitioners working with children, and investigate ways to support children with difficulties in working memory.

## Introduction

Working memory is defined as the ability to store and process a limited amount of information over short time periods to support ongoing cognitive activities [1,2]. Working memory is also part of the broader construct of 'executive function'; an umbrella term that encompasses the processes responsible for purposeful and goal-directed behaviour [3,4].

Working memory is essential for successful engagement in classroom activities [5], including the ability to remember and follow directions and instructions, and to engage effectively with problem-solving [6,7]. In mathematics, working memory is required to hold number combinations in mind [8] and when reading, working memory is required to keep relevant speech sounds in mind, match them up with corresponding letters, and then combine them to read words [9,10]. Indeed, working memory is positively associated with improved performance on school-based tests of English, Mathematics, and Science [11–13], and meta-analyses have found associations between working memory and mathematical performance [14,15], broad reading abilities [16], and reading comprehension ability [17]. In addition, working memory ability may underlie many broader cognitive abilities [18].

Given the importance of working memory for children's learning and educational attainment, it is vital to understand how working memory works and what factors might influence its development. One such factor is socioeconomic position, referring to the social and economic factors that influence the positions individuals or groups hold within the structure of a society [19]. Socioeconomic position has been shown to influence multiple developmental outcomes, including educational attainment [20]. Socioeconomic gaps are also evident in children's receptive language, general locomotor skills, and general cognitive abilities as early as 22 months old [21,22] and in school readiness, verbal ability, and spatial ability at ages 3 and 5 [23]. Ethnicity is also an important factor to consider within the context of socioeconomic position and developmental outcomes, as minority ethnic groups tend to experience both lower levels of socioeconomic position [24] and educational attainment [25]. Given the very strong associations that working memory has with broad cognitive abilities [18], and with educational attainment [11–17], it may provide a potential pathway for understanding socioeconomic inequalities in children's outcomes. A better understanding of the association between socioeconomic position and working memory may therefore provide an understanding of one of the pathways by which socioeconomic disadvantage negatively impacts upon children's educational attainments, and a potential route for reducing socioeconomic inequality.

Socioeconomic disadvantage is hypothesised to influence child outcomes negatively through experiences of stress and lack of access to resources [26,27]. For example, early childhood poverty is associated with increased allostatic load, a measure of physiological stress [28], and lower family income is associated with lower levels of cognitive stimulation in the home environment [29]. Whilst socioeconomic disadvantage may *negatively* influence child development via these suggested mechanisms, it may also be the case that enhanced social position means provides more *positive* enrichment and opportunities–resulting in better child development [30]. Indeed, transactional models posit that associations between genes, cognitive development, and academic outcomes are more strongly related in more advantaged socioeconomic circumstances [31,32]. Socioeconomic position could therefore influence children's working memory via any or all of these mechanisms.

However, there is disagreement about whether or not socioeconomic disadvantage does affect working memory. Some studies have shown no link between socioeconomic position and working memory [33–35]. In contrast, other studies have found that socioeconomic disadvantage is associated with significant impairments in working memory ability [36–38]. The need to understand the precise association between socioeconomic position and working memory and the lack of consensus regarding this association indicates that a systematic review and meta-analysis of relevant studies is necessary.

No study has systematically synthesised the literature investigating the association between child socioeconomic disadvantage and working memory. Lawson, Hook and Farah [2018] investigated the association between executive functions and socioeconomic status, finding a significant, but small, association across 25 studies [39]. They found that the association was not moderated by the age of the sample or the socioeconomic indicator used. Regarding working memory specifically, Lawson et al [2018] did briefly report some follow-up analyses looking at the different components of executive function using a sub-set of twelve studies, and found a similar level of association between socioeconomic position and working memory. However, they only included studies reporting a Pearson's *r* correlation. Studies that investigate socioeconomic differences in working memory often categorise children into "high" and "low" socioeconomic groups, meaning that many relevant studies would be excluded from this analysis.

Further, Lawson et al. [2018] combined all measures of working memory into one summary score for their meta-analysis. A prominent model of working memory, the Multicomponent Model, differentiates between different components of working memory, with a central executive that acts as an attentional control system, and two sub-systems that act as simple storage components [1,40]. These two storage components represent different modalities, where the phonological loop stores verbal information and the visuospatial sketchpad stores visual, spatial, and haptic information. Other influential models see working memory as a more unified construct and do not support the idea of separable, specialised working memory components [41,42]. However, many standardised measures of working memory (e.g., the Working Memory Test Battery for Children–[43]) contain tasks that differ by modality presentation (e.g., verbal vs. visuospatial stimuli) or by whether the information needs simply to be stored (e.g., forward digit span) or to be manipulated in some way (e.g, backward digit span). Further, neurocognitive studies have indicated that activities requiring attentional control are more uniquely associated with brain activation in the prefrontal cortex [44], whereas passive storage is related to activation within different networks, such as Broca's and Wernicke's area and the right hemisphere [45].

Given that socioeconomic disadvantage has been shown to have different patterns of association with particular aspects of cognition [33,38] and that distinct components of working memory are associated with specific underlying neurological structures, socioeconomic disadvantage may also have specific associations within distinct components of working memory.

Indeed, previous studies have shown that the magnitude of the association between socioeconomic disadvantage and working memory does change dependent on whether the task material is verbal or visuospatial [34,46], and how working memory capacity is measured [47]. Further, some researchers have argued that simple storage may be more reliant on knowledge structures, which in turn are related to crystallized intelligence, and therefore may be more sensitive to the effects of socioeconomic disadvantage than attentional control, which is related more to fluid intelligence [10]. An increased understanding of the association between socioeconomic position and working memory, and how that association might vary between different aspects of working memory, is important for informing educational and clinical practitioners working with children from disadvantaged backgrounds. If poor working memory skills are part of the pathway by which socioeconomic position affects learning and academic attainment, then building in support mechanisms that specifically target this problem will be beneficial and contribute to improving outcomes for these children [8].

The current review therefore investigates the relationship between socioeconomic position and different components of working memory in children aged up to 18 years across a large number of studies using as wide a range of outcome variables as possible, and reports both the magnitude and the variability of these associations. For the purposes of this review we hereafter refer to functions within working memory relating to the processing of information requiring attentional control or executive control as 'complex working memory', and to functions that reflect passive storage of information as 'simple working memory' [48].

## Methods

### Protocol, registration, and reporting standards

The review protocol is registered on PROSPERO (CRD number: CRD42019134936). We used the PRISMA checklist to ensure complete and transparent reporting of methods in this review (available in supplementary online materials 4) [49].

### Study eligibility criteria

**Population, exposure, and outcome.** We used the Population, Exposure, Outcome (PEO) framework to design the inclusion criteria [50]. The population were typically developing children aged 0–18 years old. The exposure was socioeconomic position (SEP), and we included studies with any indicators of it (e.g. parental occupation, parental education, family income, area deprivation, subjective assessment of wealth, etc.). The outcome was working memory performance, defined as any behavioural task that quantified a child's working memory performance (e.g. Forwards Digit Recall, Backwards Digit Recall, Counting Recall, any 'two-back' task, Corsi). As the outcome was only working memory, and not any other cognitive or executive function tasks, studies that only reported a combined composite score on executive function were not eligible.

**Study designs.** Studies were eligible for inclusion if they used any observational design (cross-sectional and longitudinal), or any intervention design if they reported socioeconomic position and working memory at baseline, prior to the intervention. Studies had to provide quantitative data on the association between socioeconomic position and working memory, so qualitative studies were excluded. Although the protocol indicated that we would conduct additional searches, many studies were identified in the initial search and so only published studies were eligible for inclusion and additional searches (e.g. of grey literature) were not attempted.

**Study inclusion criteria.** Studies were included if they met all of the following criteria: (a) they provided data on any indicator of socioeconomic disadvantage, (b) they reported

disadvantage at the individual or group level, and compared individuals or groups on that measure of disadvantage, (c) they measured performance on at least one behavioural task of working memory and reported the results quantitatively, (d) they reported data for typically developing children aged between 0–18, (e) the study was reported in the English language, (f) the study was of any observational design, or baseline characteristics if an intervention, and (g) the study was published in a peer reviewed journal.

## Search and selection procedures

We searched Embase, Psycinfo and MEDLINE via Ovid to identify published articles from database inception until 3rd June 2021. The search strategy combined key terms with a search filter that used the PROGRESS acronym to filter for equity-focused studies [51,52]. The filter is validated in Embase and MEDLINE, and was translated for use in PsycInfo. The equity filter was combined with terms and subject headings to identify 'working memory' abilities in 'children'. The basic search strategy was: (search filter for equity studies) *AND* (subject headings) OR ("working memory".ti,ab. OR "executive function*".ti,ab. OR "short?term memory.ti,ab.") *AND* (subject headings) OR (child* OR infant OR school child* OR adolescen* OR preschool* OR pre-school* OR boy* OR girl* OR young people OR teenager* OR teen* OR youth*.mp.). The full search strategy for Embase is provided in supplementary online materials 1.

## Data extraction

The following data was extracted using a previously piloted data extraction form: location of the study, number of participants, and participant sociodemographics (gender, age range and mean, and ethnicity), exposure details (the indicator of SEP), and the measurement of the outcome (working memory). Information was extracted on ethnicity since it can be associated with socioeconomic position [24]. If information regarding participant ethnicity was not available, then data on language spoken was extracted instead if it was provided. If a study did not report information for any of the details, this was marked "NR" (not reported).

## Risk of bias in individual studies

Risk of bias was assessed using one of two tools: cross-sectional studies were assessed using the AXIS appraisal tool [53], and longitudinal studies were assessed using The National Institutes of Health (NIH) Quality Assessment Tool for Observational Cohort and Cross-Sectional Studies [54]. Risk of bias was assessed at the study level. Particular attention was paid to three key factors in both tools: (1) The selection of a defined target population with reference to the population's socio-demographics, with a detailed sampling frame and selection process; (2) the measurement or consideration of screening to categorise children as 'typically developing' in the inclusion criteria for the study; (3) the measurement of working memory using a validated and or referenced task.

  If a study met all three of these conditions, and successfully met the majority of the criteria from the relevant quality assessment tool, it was labelled as low risk of bias. Studies that did not meet any of the above three conditions were labelled as high risk of bias. If a study only met one or two of the three conditions, then the context of the study and other criteria within each risk of bias tool were considered before risk of bias was assigned.

## Validity and reliability of review

The first reviewer (KM) screened all eligible abstracts and full texts; and a second (MB) screened a random 10% of excluded abstracts and full texts. KM extracted all data and then

MB checked the data extraction and risk of bias assessments for 50% of all included studies. Agreement between the reviewers was considered to be acceptable if it was at least 90% at all stages, and any disagreements were resolved through discussion.

## Data synthesis

We used two methods of numerical data synthesis: random-effects meta-analyses to combine studies with eligible effect estimates, and a Harvest plot to synthesise findings from otherwise eligible studies without effect size estimates or that used a composite measure of working memory. Fig 1 provides a summary of how studies were selected into each data synthesis method.

Two meta-analyses were conducted by the type of working memory: (1) simple working memory and (2) complex working memory. Studies were therefore included in a meta-analysis if they reported a useable (or convertible) unadjusted effect size between socioeconomic position and working memory on $\geq 1$ task(s) of working memory that could be conceptualised as either simple working memory, or complex working memory. We also conducted subgroup estimation within both meta-analyses, depending on whether the task modality was verbal or visuospatial. A small number of studies combined verbal and visuospatial task modalities and in order to include as many studies as possible within this analysis, we still included those studies that had a combined score, as long as they had separate measurements of simple working memory and complex working memory. Cohen's *d* effect sizes were calculated for all studies that provided mean scores across two groups of SEP. Not all studies provided mean scores and we converted correlations to Cohen's *d* effect size where possible, using formulae provided by Borenstein et al., (2009) [55]. This therefore means that the meta-analysis represents a metric comparing lower socioeconomic position to higher socioeconomic position groups.

The Harvest plot contained studies that were deemed ineligible for the meta-analysis. A small number ($n = 8$) of studies used composite measures of working memory (with both simple and complex working memory tasks). We decided to include these for the sake of completeness, even though one of our key aims was to investigate the association between socioeconomic position and different components of working memory. The Harvest plot

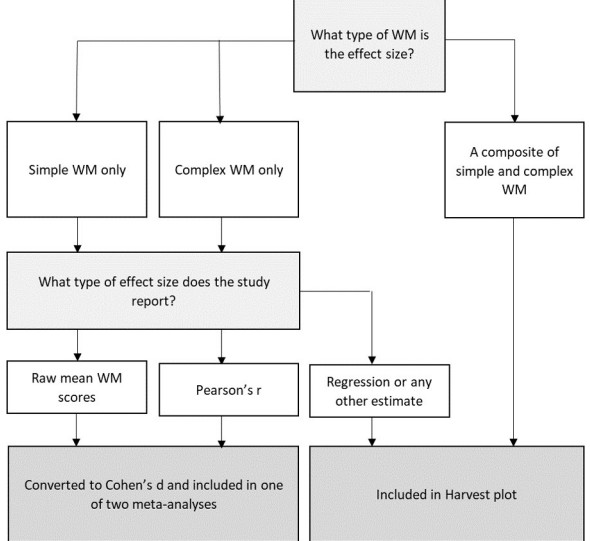

**Fig 1. Selection process and eligibility for inclusion in meta-analyses or Harvest plot.**

therefore included studies that reported what we categorised as a composite working memory score: (i) either a combined score on tasks of *both* ≥1 simple working memory task(s) and ≥1 complex working memory task(s) or (ii) a single task of working memory including elements of both simple and complex working memory. Further, we also included studies which reported (iii) an effect size that was adjusted for other factors and so could not be appropriately used in meta-analysis (e.g. regression analysis).

### Data synthesis: Meta-analytic methods

**Investigation of heterogeneity.** Heterogeneity was calculated for each meta-analysis using the $I^2$ statistic, and 95% prediction intervals. The $I^2$ statistic is a statistical test where 0% to 40% might represent unimportant heterogeneity, 30% to 60% moderate heterogeneity, 50% to 90% substantial heterogeneity, and 75% to 100% considerable heterogeneity [56]. Prediction intervals present the heterogeneity in the same metric as the original effect size measure and illustrate the range of true effects that can be expected in future settings [57]. To investigate sources of heterogeneity, we also undertook sensitivity analyses and meta-regression analyses.

**Sensitivity analyses.** The majority of studies reported two or more effect sizes that were eligible for the meta-analyses (70%), e.g. SEP-working memory correlations for the same individuals at different ages or time points [58], and these estimates are statistically dependent. We first averaged the effect sizes to give one effect size per study, however, this may result in a loss of potentially important information, improper sampling variance, or a higher probability of type-2 errors [59–61]. As a sensitivity analysis we re-estimated the meta-analyses using the Robust Variance Estimation (RVE) method, which accounts for statistically dependent effect sizes [62], and compared the results to the main analyses where effect sizes were averaged. As RVE only provides an overall summary effect size, neither subgroup analysis by verbal and visuospatial or estimation of heterogeneity parameters were possible for this sensitivity analysis. We also conducted a sensitivity analysis to examine the potential effect the inclusion or exclusion of a single study with a very large effect size.

**Meta regression analyses.** Meta-regression allows the effect of both continuous and categorical characteristics on the estimated effect size to be investigated. It estimates whether the association of interest (socioeconomic position and working memory) is associated with an investigated characteristic, where a significant *p* value indicates evidence that it is a significant moderator [63].

We tested moderation of the association between socioeconomic position and working memory with three characteristics as pre-specified on PROSPERO: (a) the type of socioeconomic indicator (whether it was a composite or single indicator), (b) the risk of bias (low or high), and (c) the task modality (verbal or visuospatial). We also tested three further post-hoc moderators; (d) the type of effect size (Cohen's *d* or converted from Pearson's *r*), (e) whether the effect size had been averaged from >1 estimate(s) or not, and (f) the mean age of the sample. We wanted to ensure the effect sizes were not affected by the way they were converted or combined. We tested moderation by age as enough data was available to do so, and to investigate if this could explain the heterogeneity found. If the statistical significance test was *p* < .05, the tested variable was considered to be a significant moderator of the association between socioeconomic position and working memory.

**Publication bias.** Publication bias was investigated for each meta-analysis using a funnel plot, with effect estimates plotted against standard errors of effect estimates, and Egger's test, which estimated whether the association between study size and effect estimates was greater than expected by chance [64]. As publication bias is typically assessed per meta-analysis, this was not done by subgroup analysis (verbal vs visuospatial).

**Software.** We used Stata-16 and the Meta command for the computation of effect sizes, calculation of heterogeneity statistics, calculation of pooled effect sizes, meta-regressions, and producing the forest-plots [65]. For the sensitivity analysis regarding robust variance estimation, we used the Stata-16 user written command *robumeta* [66].

### Data synthesis: Harvest plot

Studies were grouped on the Harvest plot based on whether reported findings illustrated a positive association (lower socioeconomic position and lower working memory), negative association (lower socioeconomic position and higher working memory), or no association. Outcome measures, study designs, and study quality were summarised [67]. The Harvest plot enables the reader to judge where the majority of studies lie in relation to competing hypotheses, and where the highest quality studies are. As some studies included numerous socioeconomic indicators, those that report multiple effect sizes are represented in the plot more than once. The columns represent effect sizes across composite working memory, simple working memory, and complex working memory.

## Results

### Study selection

The study selection process is reported according to the PRISMA STATEMENT (http://www.prisma-statement.org/) diagram.

Fig 2 shows the selection process for all included studies, a total of 64 studies were eligible for the review. The majority of these were meta-analysed (n = 36), and the remaining studies were included in the Harvest plot (n = 28).

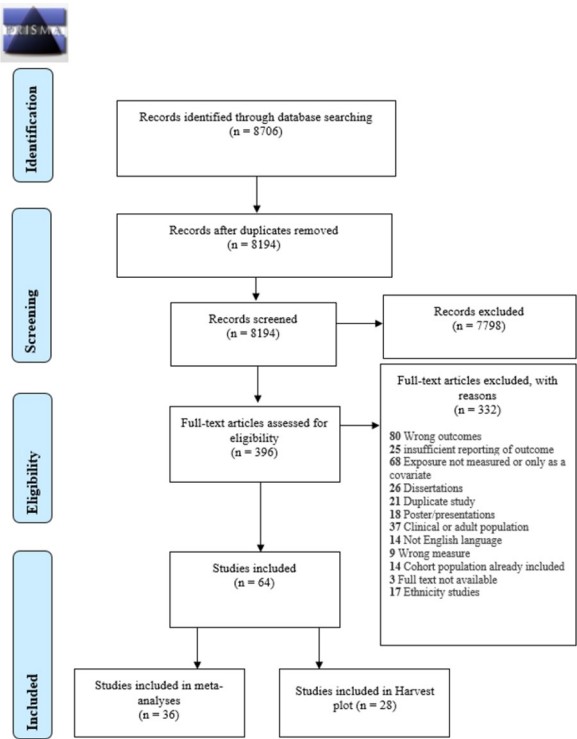

**Fig 2. PRISMA 2009 flow diagram for all included studies.**

## Study characteristics

A summary of included study characteristics and risk of bias results are provided in online supplementary materials 2, and a full reference list of studies is provided in online supplementary materials 3. The 64 studies were conducted across 17 different countries, with the most frequently studied populations being the USA (34%) and Brazil (11%). The age range included children from aged 1 month to 17 years, and the majority of studies included only children under 10 years old (56%), whilst the remaining studies included a mix of ages both below and above 10 years old (44%). A variety of measures of socioeconomic position were used, with most studies including traditional measures of parental occupation, parental education and family income (80%). Other studies used measures including school socioeconomic coefficients, country specific socioeconomic indicators, and neighbourhood measures (15%), and a few studies used subjective measures of social status (5%). Many of the studies included ≥2 ethnic groups, with both ethnic majority and minority children (45%). The remaining studies either included only ethnic majority children (20%), only ethnic minority children (3%), or either did not report ethnicity at all or only reported the language spoken by the sample (30%). The majority of studies (80%) were rated as low risk of bias; studies at high risk of bias (20%) were rated as high risk either because they did not specify the socio-demographics of their population, did not have clear inclusion criteria, or used a measure of working memory that was not referenced or validated.

## Meta-analyses

**Summary of effects.** There were 25,249 individual participants from 35 individual studies included across both meta-analyses. Results are presented firstly for the meta-analysis of simple working memory and then the meta-analysis of complex working memory. Within each meta-analysis, the subgroup analysis is presented by modality (verbal vs visuospatial).

*Simple working memory.* Fig 3 shows the meta-analysis of simple working memory, which included 27 studies with 14,328 participants (including 7006 from one study). The effect size and 95% CI was 0.45 (0.27 to 0.63). In the task modality subgroup analysis, the verbal estimate and its 95% CI was 0.47 (0.15 to 0.79), the visuospatial estimate 0.40 (0.23 to 0.57), and the combination of verbal and visuospatial estimate was 0.55 (0.16 to 0.94).

*Complex working memory.* Fig 4 shows the complex working memory meta-analysis, which included 23 studies with 20,651 participants (including 14,000 from one study). The effect size and 95% CI was 0.52 (0.31 to 0.72). In the subgroup analysis of task modality, the verbal estimate was 0.54 (0.25 to 0.83), the visuospatial estimate 0.41 (0.13 to 0.69), and the combination of verbal and visuospatial estimate 0.62 (0.42 to 0.82).

**Heterogeneity.** Heterogeneity was high overall. $I^2$ was 85% overall in simple working memory, with substantially higher heterogeneity in simple verbal working memory (89%) than simple visuospatial working memory (48%). $I^2$ was 87% overall in complex working memory, with again substantially higher heterogeneity in complex verbal working memory (91%) than complex visuospatial working memory (47%) (likely due to the subgroup analysis including only 4 studies). Prediction intervals were wide and overlapped with the null, indicating some uncertainty about the direction and magnitude of any effect to be expected in a new study. The 95% prediction intervals were -0.399 to 1.297 for simple working memory, and -0.407 to 1.438 for complex working memory.

**Publication bias.** We assessed publication bias for each of the two meta-analyses. The funnel plots in Figs 5 and 6 were both judged to be symmetrical and did not show an association between study size and study effect estimates. The Egger's tests were both non-significant

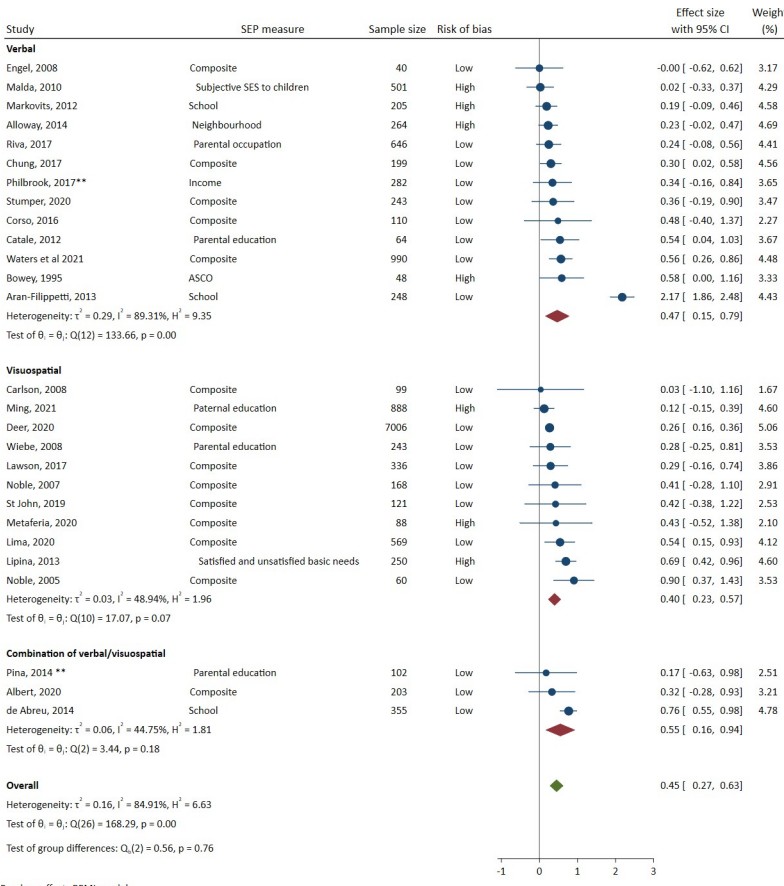

**Fig 3. Meta-analysis of the association between socioeconomic position and simple working memory (sorted by effect size).** Note: A double asterisk ** indicates a cohort or longitudinal study. Effect sizes to the right of the 0 line favour the higher socioeconomic positioned groups.

(simple working memory $p = .44$, complex working memory $p = .93$), again indicating low risk of publication bias.

**Sensitivity analysis.** The RVE analysis of simple working memory included 27 studies with 50 individual effect sizes. The simple working memory effect size and 95% CI was 0.44 (0.24 to 0.64). The RVE analysis of complex working memory included 23 studies with 39 individual effect sizes. The complex working memory effect size and 95% CI was 0.53 (0.30 to 0.75). As these estimates are extremely similar to when effect sizes were averaged within studies, here we have presented only the forest plots for the averaged effect sizes and the discussion focuses on the results when the effect sizes were averaged.

Removing one study [68] with substantially larger effect sizes than others in both meta-analyses ($d = 2.17$ in simple working memory, and $d = 2.22$ in complex working memory) reduced the effect sizes by approximately 0.1 (simple working memory from 0.45 to 0.37 and complex working memory 0.52 to 0.42). Removing this study also substantially reduced the heterogeneity as measured by $I^2$, from 87% to 48% in simple working memory, and from 88% to 43% in complex working memory. As the overall effect sizes were still within the bounds interpreted as "medium", we retained the Aran-Filippetti (2013) study in all meta-analyses.

**Meta regression analyses.** Results from the meta-regression analysis are presented in Table 1. We conducted pre-specified moderation analyses by the task modality, and type of

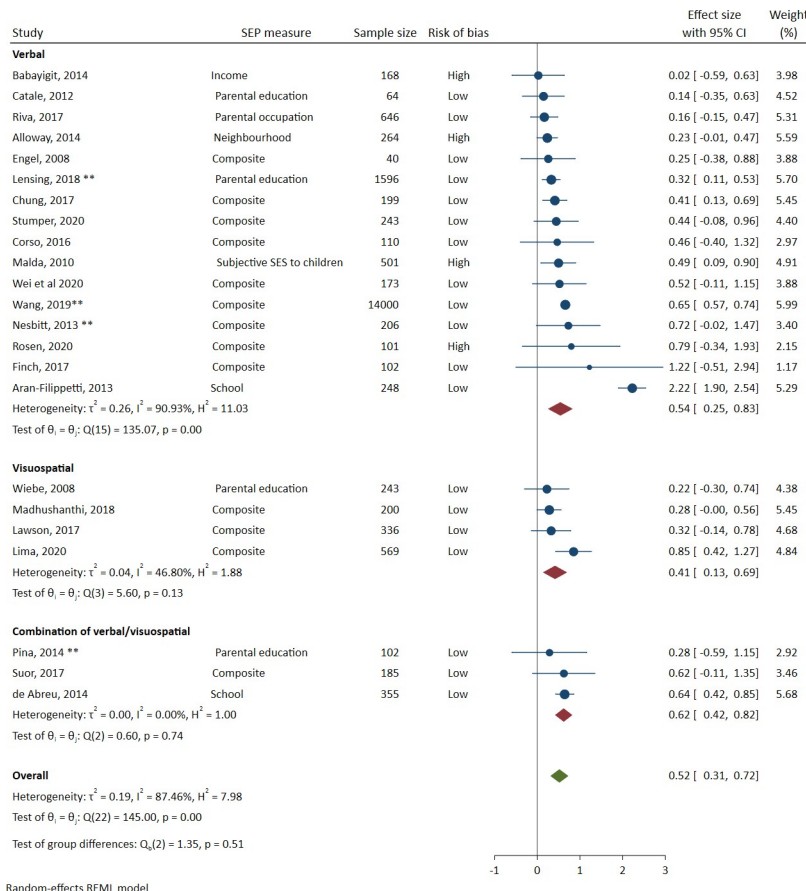

**Fig 4. Meta-analysis of the association between socioeconomic position and complex working memory (sorted by effect size).** Note: A double asterisk ** indicates a cohort or longitudinal study. Effect sizes to the right of the 0 line favour the higher socioeconomic positioned groups.

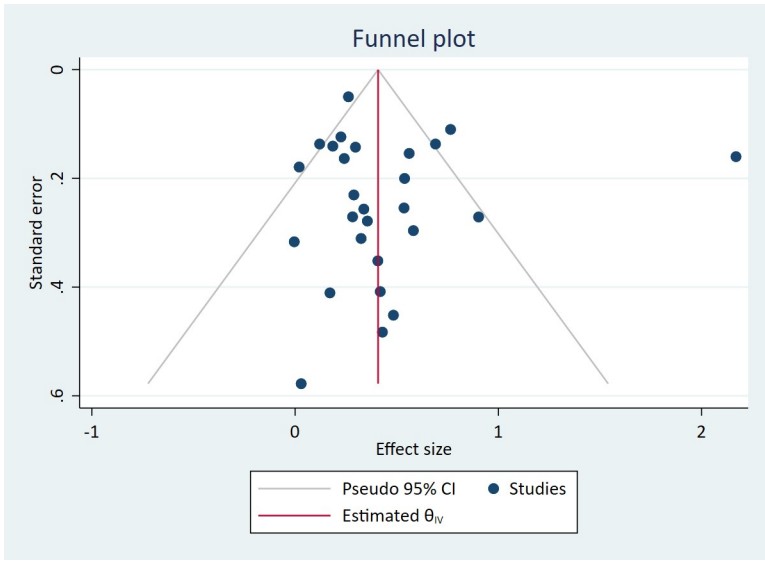

**Fig 5. Funnel plot for simple working memory meta-analyses.**

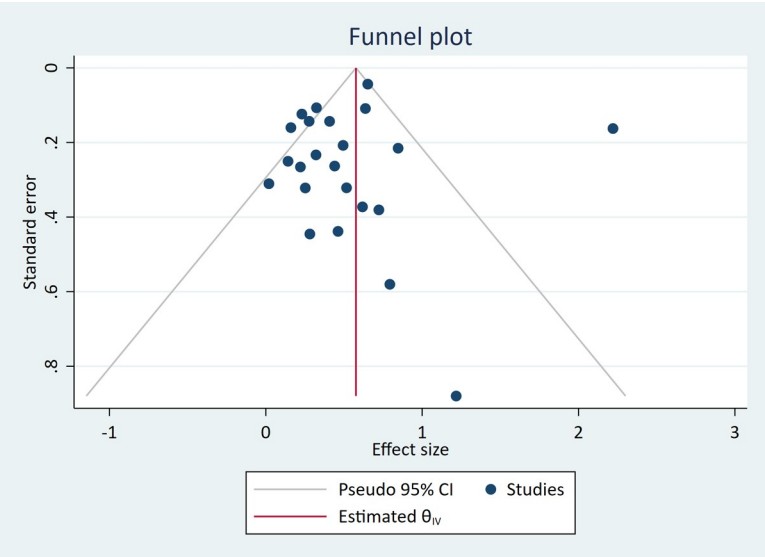

**Fig 6. Funnel plot for complex working memory meta-analyses.**

socioeconomic indicator; however, neither of these variables significantly moderated the association between socioeconomic position and working memory. As a post-hoc analysis, we found that age in years did not significantly moderate the association (for those studies that reported mean age in years), nor did the risk of bias of the study. We also found that whether the effect size was averaged or not did not significantly moderate the association, nor did whether the effect size was converted from Pearson's r. However, the test was borderline significant ($p = .05$) for the second moderation test in simple working memory.

## Harvest plot

There were 28 studies included in the Harvest plot using 51 effect sizes from 12,488 individual participants. The majority of studies contributed $\geq 2$ effect sizes (58%).

**Table 1. Meta-regression analyses results.**

| | Simple working memory | | Complex working memory | |
|---|---|---|---|---|
| **Regression factor** | **B (95% CI)** | ***p*** | **B (95% CI)** | ***p*** |
| **Pre-specified** | | | | |
| Task modality (0 = verbal, 1 = visuospatial)* | -0.07 (-0.50 to 0.35) | .47 | -0.26 (-0.91 to 0.38) | .42 |
| Socioeconomic indicator (0 = single, 1 = composite) | -.11 (-0.48 to 0.27) | .58 | -.00 (-0.44 to 0.43) | .97 |
| **Post-hoc** | | | | |
| Risk of bias (0 = low risk, 1 = high risk) | -0.20 (-.60 to .21) | .36 | -0.20 (-0.77 to 0.36) | .49 |
| Effect size (0 = Cohen's d, 1 = Converted from Pearson's r) | -0.35 (-0.71 to -0.00) | .05 | -0.18 (-0.63 to 0.26) | .41 |
| Effect size (0 = single, 1 = averaged) | -.17 (-0.55 to 0.02) | .40 | 0.19 (-0.27 to 0.65) | .42 |
| Age in years** | -.05 (-0.11 to .00) | .09 | -.02 (-.07 to .03) | .43 |

*Three studies used combined estimates of verbal and visuospatial task modalities, and were excluded from this analysis.

*Nine studies did not report a mean age of their sample, and were excluded from this analysis.

In Fig 7, The Harvest plot shows the distribution of statistically significant associations and non-statistically significant associations across composite working memory, simple working memory, and complex working memory by socioeconomic position measure. Studies only showed a positive association (increased socioeconomic position and increased working memory), or no association, so there is no column representing negative association. The abundance of studies in the composite working memory columns relative to the simple and complex working memory columns reflects that studies with composite working memory measures were not included in the meta-analyses.

Nineteen individual studies including 7826 participants provided 43 effect sizes on composite working memory. The majority of studies found composite working memory to be significantly positively associated with composite socioeconomic indicators, household wealth and parental education, and most of these studies were rated as low risk of bias. Two studies rated as low risk of bias found no association between composite socioeconomic position and working memory, and two studies rated as low risk of bias found single parent status to not be associated with verbal working memory. Eight individual studies including 7826 participants provided 14 effect sizes for simple working memory. Simple working memory was found to be associated with composite socioeconomic position indicators, household wealth, and parental education, and most of these studies were rated as low risk of bias. Only three studies found no association between socioeconomic position and simple working memory. Three individual studies including 641 participants provided four effect sizes for complex working memory. Complex working memory was found to be associated with composite socioeconomic position and household wealth in two different studies, one of which was rated as low risk of bias. The third study of

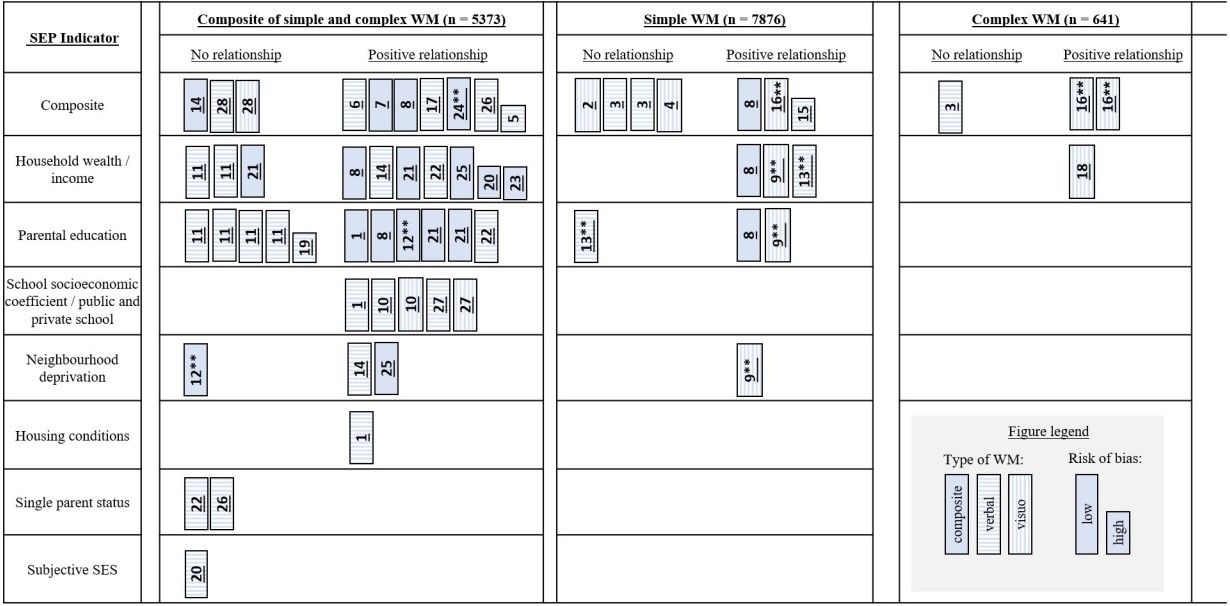

**Fig 7. Harvest plot of the association between different socioeconomic position indicators with composite working memory, simple working memory, and complex working memory.** Note: Study IDs are indicated on each bar as follows: 1. Aran-Filippetti & Richard De Minzi, 2012; 2. Brito et al., 2021; 3. Cockcroft, 2016; 4. Daubert and Ramani, 2020; 5. Dicataldo and Roch, 2020; 6. Dilworth-Bart, 2012; 7. Farah et al. 2006; 8. Fernald et al., 2011; 9. Flouri et al., 2019; 10. Guerra et al., 2020; 11. Hou et al., 2020; 12. Hackman et al. 2014**; 13. Hackman et al. 2015**; 14. He and Yin, 2016; 15. Jacobsen et al. 2017;16. Kobrosly et al. 2011; 17. Korecky-Kroll et al., 2019; 18. Leonard et al. 2015; 19. Maguire and Schneider, 2019; 20. Miconi et al. 2019; 21. Murtaza et al., 2019; 22. Passareli-Carrazzoni et al. 2018; 23. Piccolo et al. 2019; 24. Rhoades, 2012**; 25. Rowe et al. 2016; 26.Sarsour et al. 2011; 27. Tine, 2014; 28. Vandenbroucke et al. 2016. The plot bar lengths indicate whether the study was at low or high risk of bias. A double asterisk ** indicates a cohort or longitudinal study.

complex working memory, rated as low risk of bias, found no association with composite socioeconomic position.

Overall, the Harvest plot indicates an association between socioeconomic position and different types of working memory across different indicators of socioeconomic position, that appear unrelated to risk of bias. Although there were some studies that found evidence against these hypotheses, the weight of evidence rated as low risk of bias was much more in favour of supporting the evidence for an association.

## Discussion

This is the first systematic review of the association between socioeconomic position and children's working memory abilities, with a very large sample of individual participants (n = 37,737) across two data synthesis methods. In a meta-analysis of 27 studies with 14,328 participants, socioeconomic position was associated with overall simple working memory ability with a medium effect size. In a meta-analysis of 23 studies with 20,651 participants, socioeconomic position was associated with overall complex working memory, also with a medium effect size. Furthermore, socioeconomic position was significantly associated with both verbal and visuospatial tasks within both simple and complex working memory. We also synthesized 28 studies including 12,488 participants with more diverse measures of effect using a Harvest plot, finding that most predictors of socioeconomic position are associated with working memory. The findings are consistent with literature that views socioeconomic disadvantage to be associated with impairments in working memory [36,37], and therefore does not support the view that working memory is unrelated to socioeconomic disadvantage [10,33].

We found that the magnitude of the association was similar across both the simple and complex working memory meta-analyses ($d = 0.45$ and $d = 0.52$ for simple and complex working memory, respectively). This indicates that child socioeconomic disadvantage is associated with not only difficulties in the simple storage of information, but also with the ability to process and manipulate information. This does not support the argument that simple working memory may be more sensitive to the effects of socioeconomic disadvantage than complex working memory due to being more reliant on knowledge structures [10].

We also investigated whether the magnitude of the association differed by modality (verbal and visuospatial), finding a similar magnitude of associations within the simple working memory meta-analysis ($d = 0.47$ and $d = 0.40$ for verbal and visuospatial, respectively) and the complex working memory meta-analysis ($d = .54$ and $d = 0.41$ for verbal and visuospatial working memory, respectively). We also tested this formally through meta-regression, and found that task modality did not moderate the association between socioeconomic position and working memory. Still, visuospatial working memory tended to have smaller effect sizes, and this is likely because our subgroup analyses of the visuospatial studies contained fewer effect sizes–perhaps due to difficulties with assessing visuospatial working memory in children. Future studies could examine the consistency of the association between socioeconomic disadvantage and visuospatial working memory to explore this further.

Modular working memory theories, such as the multicomponent model [1,69], propose separate components for different functions within working memory. In contrast, unitary approaches such as the attentional control model [70] and Cowan's embedded processes model [41,71] do not support dissociable components. Our results showed a similar level of association between socioeconomic position and different components of working memory. This could be seen as evidence to support a more unitary approach to working memory. However, it may also be that the separate components of working memory are affected by socioeconomic position to a similar extent.

The findings indicated significant heterogeneity across the studies, with prediction intervals crossing the null line. However, the prediction intervals included a high upper boundary and the average effect size in both studies is medium, indicating that a significant average effect is likely to exist in future settings [57]. High heterogeneity can be due to clinical or methodological diversity, and in most cases, it is likely due to both [72]. It was difficult to ascertain the source of heterogeneity in this review as we synthesised a large number of studies, varying in both methodological and participant characteristics. The finding of high heterogeneity can be interpreted as an indication that the association between socioeconomic position and working memory is highly likely to vary across different settings and participants. Further, the prediction intervals overlapped with the null, indicating some uncertainty about the direction and magnitude, and therefore uncertainty regarding the generalizability of the effect to future studies.

We investigated some sources of the high heterogeneity through exploration of potential moderating characteristics using meta-regression [73]. We found that risk of bias did not moderate the association, where studies at high risk of bias had similar associations as those with low risk of bias. This may be because only a small proportion of meta-analyzed studies were assessed to be high risk of bias (20%). We found that child age did not moderate the association which suggests that socioeconomic disadvantage is detrimental to children's working memory regardless of child age, and does not accumulate throughout childhood. Still, as the majority of studies in this review were cross-sectional in design, this finding warrants further validation with longitudinal studies. Finally, whether or not the socioeconomic indicator was a single item or a composite did not moderate the association. We were only able to compare the difference between single and composite indicators of socioeconomic position, as there were not enough data to explore differences between different individual indicators. This finding therefore warrants further exploration across single indicators of socioeconomic position and working memory, as this may give more insight into any causal mechanisms between disadvantage and working memory.

We also explored the influence of our data synthesis methods on the effect sizes through meta-regression. We found that the association was not moderated by whether the effect size had been averaged or not (and this was further confirmed with the RVE sensitivity analysis). Finally, there was some evidence to suggest the association between socioeconomic position and simple working memory was moderated by whether effect sizes had been converted from Pearson's $r$, with smaller effect sizes for those that had been converted ($B = -0.35$, $p = .05$), although this finding did not hold for complex working memory. Studies that had been converted from Pearson's $r$ showing some evidence of weaker associations than those that used mean scores across two groups is relatively unsurprising since studies representing a continuum of socioeconomic position would have weaker associations than those comparing two extreme groups of socioeconomic position. This finding may therefore suggest the true association between two extreme groups of socioeconomic position and working memory is even larger than we have estimated here.

The finding that working memory is associated with socioeconomic position has important implications for educational and clinical professionals who work with children from disadvantaged backgrounds. Poor working memory ability is strongly linked to worse learning and educational outcomes. Therefore, practitioners should consider whether these children could benefit from an environment that actively scaffolds and supports children's working memory [8].

## Directions for future research

We did not systematically investigate causal or contextual factors that may mediate the association between socioeconomic position and working memory as this was not the focus of our

review. Further investigation using longitudinal studies would enable exploration of the complex interplay between different factors and the effect on working memory, and we recommend some factors for future research here.

One potential moderating characteristic is ethnicity. It was not possible for us to explore ethnicity as a moderator as nearly half of the studies in this review included two or more ethnic groups, with both ethnic majority and minority children. Minority ethnic groups tend to experience higher levels of socioeconomic disadvantage [24], and it has previously been found that socioeconomic disadvantage is associated with worse working memory in ethnic minority children, whilst ethnic majority children at different levels of socioeconomic risks have similar working memory ability [74]. This therefore indicates that a difference across ethnic groups in how socioeconomic position may influence working memory may exist, and ethnicity may be a potential moderator of this association. The disadvantage faced by ethnic minority groups may exacerbate the negative association between socioeconomic position and working memory, something that could be explored more fully in future research.

As mentioned in the introduction, two key potential mediating causal factors between socioeconomic disadvantage and child development are the home learning environment and chronic stress [27–29]. Socioeconomic disadvantage may impair parents' ability to provide home enrichment resources and activities (use of toys, books, and learning experiences), which has been found to be associated with children's working memory [36]. Additionally, allostatic load, a biological marker of cumulative chronic stress, has been found to mediate the associations between childhood poverty and adult working memory ability [75], and this is consistent with a systematic review that found an association between early life stress and working memory [76].

A contextual factor that may induce differences is 'stereotype threat'. Stereotype threat occurs when people are, or feel themselves to be, at risk of conforming to stereotypes about their own social group, and has been discussed as a contributing factor to the achievement gap between children of low and high socioeconomic status [77] and children from different ethnic groups [78]. Schmader, Johns and Forbes (2008) theorise that for those at risk of being negatively stereotyped about their abilities, stereotype threat increases physiological stress at the time of testing, active monitoring of performance, and efforts to suppress negative thoughts. These physiological and psychological mechanisms consume executive resources needed to perform well on cognitive tasks, including tasks of working memory [79]. Whilst the majority of studies investigating stereotype threat explicitly prime stereotypes prior to test tasking [77,80], children may still be aware of their disadvantage in a test setting without explicit priming. As socioeconomically disadvantaged children become aware of their relative disadvantage early in life [81], it seems plausible that stereotype threat may underpin *some* socioeconomic differences in working memory.

Finally, whether the association between socioeconomic position and working memory has a particular impact on specific areas of educational attainment is of interest, and longitudinal analyses examining working memory as a mediator between socioeconomic position and educational attainment could reveal more about these associations. Indeed, one study has found that executive function partially mediates the association between socioeconomic position and math achievement [37].

## Strengths and limitations

This systematic review included a broad range of studies using a variety of methods to assess the association between socioeconomic position and working memory. The use of a comprehensive search strategy utilising the equity filter based on PROGRESS [51] allowed us to

identify a large number of studies (>7000 at the initial stage). Unlike previous reviews on this topic, inclusion was not constrained to any particular estimation method, but included all studies with any quantitative measure of association between socioeconomic position and working memory. The use of the Harvest plot allowed us to include studies using any estimation method and reduces the likelihood of bias in the findings. This systematic review is the first to analyse the association between socioeconomic position working memory, and explores the association by the different components of working memory. The separation of the results into the different components of working memory allows the results to be applicable to both modular and unitary working memory models, as the summary effect sizes for each component can be considered to reflect those different components of working memory, or they can be combined to consider working memory as one construct.

As the majority of studies used cross-sectional designs, we were not able to establish causality from the associations reported in this review. However, we have highlighted potential causal factors for future studies to investigate. We converted effect size measures to a common metric, and thus the conversion into Cohen's $d$ therefore means that our meta-analyses analysed socioeconomic position as a dichotomous variable with two groups of socioeconomic position–which is not how socioeconomic position is actually distributed. However, the alternative would have been to exclude the studies that happened to use an alternate metric— potentially resulting in a biased sample of studies [73].

## Conclusion

To conclude, this is the first systematic review specifically to investigate the association between socioeconomic disadvantage and working memory, and to analyse that relationship across different components of working memory. The results showed that socioeconomic disadvantage was associated with lower working memory ability in children, and that this association was similar across different working memory tasks. This review adds to a large body of evidence demonstrating the unjust developmental inequalities faced by children from socio-economically disadvantaged families face [20–23]. Given the strong association between working memory, learning, and academic attainment, there is a definite need to investigate whether the pathway between socioeconomic position and working memory may explain some of the stark socioeconomic inequalities in children's educational attainments. In addition, there is a need to share these findings with practitioners working with children, and to continue to investigate ways to support children with difficulties in working memory.

## Supporting information

**S1 File. Search strategy for Embase.**
(DOCX)

**S2 File. Table of study characteristics.**
(DOCX)

**S3 File. References for studies included in systematic review.**
(DOCX)

**S4 File. PRISMA 2009 checklist.**
(DOC)

## Author Contributions

**Conceptualization:** Kate E. Mooney, Kate E. Pickett, Amanda H. Waterman.

**Formal analysis:** Kate E. Mooney.

**Investigation:** Kate E. Mooney.

**Methodology:** Kate E. Mooney, Stephanie L. Prady, Kate E. Pickett, Amanda H. Waterman.

**Project administration:** Kate E. Mooney, Mary M. Barker.

**Supervision:** Stephanie L. Prady, Kate E. Pickett, Amanda H. Waterman.

**Writing – original draft:** Kate E. Mooney.

**Writing – review & editing:** Kate E. Mooney, Stephanie L. Prady, Mary M. Barker, Kate E. Pickett, Amanda H. Waterman.

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
