## [Decision Letter · Decision Letter 0]

7 Oct 2021

PONE-D-21-23197The association between socioeconomic disadvantage and children’s working memory abilities: A systematic review and meta-analysisPLOS ONE

Dear,

Thank you for submitting your manuscript to PLOS ONE. After careful consideration, we feel that it has merit but does not fully meet PLOS ONE’s publication criteria as it currently stands. Therefore, we invite you to submit a revised version of the manuscript that addresses the points raised during the review process.

 Please submit your revised manuscript by 1st Novermber 2021. If you will need more time than this to complete your revisions, please reply to this message or contact the journal office at plosone@plos.org. Please include the following items when submitting your revised manuscript:A rebuttal letter that responds to each point raised by the academic editor and reviewer(s). You should upload this letter as a separate file labeled 'Response to Reviewers'.A marked-up copy of your manuscript that highlights changes made to the original version. You should upload this as a separate file labeled 'Revised Manuscript with Track Changes'.An unmarked version of your revised paper without tracked changes. You should upload this as a separate file labeled 'Manuscript'.

We look forward to receiving your revised manuscript.

Kind regards,

Muhammad Shahzad Aslam, Ph.D.,M.Phil., Pharm-D

Academic Editor

PLOS ONE

Journal Requirements:

2. Please include in the Ethics statement in the Methods section the information that the IRB waived the need for informed consent from participants

3. We note that this manuscript is a systematic review or meta-analysis; our author guidelines therefore require that you use PRISMA guidance to help improve reporting quality of this type of study. Please upload copies of the completed PRISMA checklist as Supporting Information with a file name “PRISMA checklist

Reviewers' comments:

Reviewer's Responses to Questions

**Comments to the Author**

1. Is the manuscript technically sound, and do the data support the conclusions?

Reviewer #1: Yes

Reviewer #2: Partly

2. Has the statistical analysis been performed appropriately and rigorously? 

Reviewer #1: Yes

Reviewer #2: No

3. Have the authors made all data underlying the findings in their manuscript fully available?

Reviewer #1: Yes

Reviewer #2: Yes

4. Is the manuscript presented in an intelligible fashion and written in standard English?

Reviewer #1: Yes

Reviewer #2: Yes

5. Review Comments to the Author

Reviewer #1: Review for PLoS One

PONE-D-21-23197

The association between socioeconomic disadvantage and children’s working memory abilities: A systematic review and meta-analysis

GENERAL COMMENTS: This is a very interesting review/meta-analysis, addressing a less-studied topic in the field of working memory – specifically, to examine the influence of socio-economic status on the development of working memory in children (magnitude and variability). The authors tackled this issue by reviewing publications of typically developing children from 0-18 years, with quantitative measures of working memory and SES, until June 2021. Meta-analyses were conducted with random effects analyses and Harvest Plots. The sample was very large, with 64 eligible studies and nearly 40,000 children aged 2 months to 18 years – the meta-analyses were conducted on half the sample. The authors found in the meta-analyses and supporting Harvest plot that reduced SES was linked to lower WM, particularly storage and information processing.

ABSTRACT COMMENTS:

• Comprehensive abstract, concise with relevant details and important conclusions about supporting the development of WM in children.

INTRODUCTION COMMENTS:

• Very clear introduction, framing the topic well, and highlighting the novelty/need for the current meta-analysis, good job!

• Minor point – page 6, from line 125 – repeated use of the word different makes sentence less fluent, advise revision.

• Good summary of aims to examine the components of working memory in line with children’s SES.

METHODS COMMENTS:

• Good use of the PEO framework

• Participants, exposure and outcomes all clearly described

• Clear study design

• Clear study inclusion criteria.

• Excellent search/selection strategy via PROGRESS

• Clear details about data extraction

• Excellent use of risk of bias tools

• Validity/reliability measures very thorough

RESULTS COMMENTS:

• Useful information about study characteristics, especially which countries

• Authors should be commended for the high number of individual participants

• Excellent summaries of heterogeneity, publication bias and sensitivity analyses

• The figures are excellent and help to illustrate the findings well.

• The summary paragraph at the end of the Harvest Plot section is very useful

• Overall the results section is excellent – clearly reported.

DISCUSSION COMMENTS:

• Excellent summary of findings – clearly interprets the wealth of data provided in the paper

• Good use of subheadings

• Good link between parents’ SES, allostatic load and variance in child working memory development

• Also the discussion about stereotype threat was intriguing

OVERALL RECOMMENDATION: An excellent paper, truly well executed and inspiring. Beyond the minor points above, this paper could be accepted in its current form.

• ACCEPT

Reviewer #2: Review of paper titled: The association between socioeconomic disadvantage and children’s working memory abilities: A systematic review and meta-analysis

Journal: PLOS ONE

Introduction

The authors cover a substantive amount of relevant literature in the introduction to explain the possible mechanisms through which socioeconomic position may affect complex working memory in children under 18 years of age. The authors further highlight the disagreement and conflicting findings within the literature regarding the association between socioeconomic position and complex working memory; this is not justification enough for undertaking the current systematic review and meta-analysis. It would be good to see the authors explain why this information is clinically relevant and what the potential impact it could have for children currently living in lower socioeconomic positions.

Methodology

The authors state that their protocol is registered on PROSPERO, however, a quick search using the CRD number cited shows no hits. See screen shot below.

Why have the authors chosen the PEO framework as opposed to the PICOS framework which would include a comparison group/population? PICOS – population, intervention/exposure, comparison, outcome, study design. Based on the full description provided under the study eligibility criteria section, this review is able to meet the conditions for using the PICOS framework.

Although the PRISMA checklist does not specifically state that study selection needs to be done by two independent reviewers, the PRISMA statement website makes reference to the Cochrane Handbook under the protocol guidelines section. In the Cochrane handbook is explicitly states that all studies from the search should be reviewed by at least two people working independently to check if studies meet the inclusion criteria. See https://training.cochrane.org/handbook/archive/v6.1/chapter-04#section-4-6-4

For this study, the second reviewer only reviewed a subset of the excluded studies (not the included studies) as well as the data extraction. In light of this, how do the authors reasonably classify this study selection as being part of a standardized systematic review process?

A meta-analysis is typically a metric defined by two relationships, however, from the description of how they conducted the meta-analysis, it’s not explicitly clear that they had two groups to compare in this meta-analysis. Did the authors compare lower SEP to higher SEP in the meta-analysis? If so, I think it would help to explicitly state so. If not, they that also needs to be made clear and also provide a justification for taking the meta-analysis down to only one metric.

In the PRISMA flow diagram it shows that 8192 studies were obtained after duplicates were removed. From this a further 7881 studies were excluded, which would leave you with a total of 311 eligible studies and not 396 as indicated on the flow diagram.

Results

The result section is presented well. Figures 3 & 4 contain forest plots for which the scale -1 – 3 is provided. It make it simpler for the reader, please can the authors indicate which side of the midline favours the relationship in question.

Discussion

The discussion merely presents a summary of the findings and doesn’t really create an argument or provide an explanations about the statistical findings.

In the recommendations for future studies the authors state that “it has previously been found that socioeconomic disadvantage is associated with worse working memory in ethnic minority children, whilst ethnic majority children at different levels of socioeconomic risks have similar working memory ability”. Are the authors suggesting that ethnicity itself, inherently affects working memory despite socioeconomic position? Further more they state that the “disadvantage faced by ethnic minority groups may therefore exacerbate the negative association between socioeconomic position and working memory”; can the authors please elaborate on what these disadvantages are?

In the final sentence of the conclusion the authors reveal the impact of the findings of this review. The potential for this information to be clinically and practically useful should be mentioned much earlier in the manuscript (i.e.: as part of the justification for doing this in the introduction).

6. PLOS authors have the option to publish the peer review history of their article (what does this mean?). If published, this will include your full peer review and any attached files.

Reviewer #1: **Yes: **Samantha J Brooks

Reviewer #2: No

---

## [Author Response · Author response to Decision Letter 0]

27 Oct 2021

We thank the reviewers for their comments and think the revisions we have made in response have strengthened it considerably. Where required, our responses to the reviewers’ suggestions are given in bold after each statement. The page/line numbers relate to where the change has been made. 

Journal

Response: We have ensured that PLOS ONE’s style requirements are met. 

2. Please include in the Ethics statement in the Methods section the information that the IRB waived the need for informed consent from participants

Response: Given this paper is a systematic review, informed consent was not applicable. For information on whether informed consent was sought for the individual studies included in the review, please see the individual studies themselves. 

3. We note that this manuscript is a systematic review or meta-analysis; our author guidelines therefore require that you use PRISMA guidance to help improve reporting quality of this type of study. Please upload copies of the completed PRISMA checklist as Supporting Information with a file name “PRISMA checklist”

Response: The PRISMA checklist was uploaded with the original file submission. We have uploaded it again with this resubmission. 

Reviewer #1

GENERAL COMMENTS: 

This is a very interesting review/meta-analysis, addressing a less-studied topic in the field of working memory – specifically, to examine the influence of socio-economic status on the development of working memory in children (magnitude and variability). The authors tackled this issue by reviewing publications of typically developing children from 0-18 years, with quantitative measures of working memory and SES, until June 2021. Meta-analyses were conducted with random effects analyses and Harvest Plots. The sample was very large, with 64 eligible studies and nearly 40,000 children aged 2 months to 18 years – the meta-analyses were conducted on half the sample. The authors found in the meta-analyses and supporting Harvest plot that reduced SES was linked to lower WM, particularly storage and information processing.

ABSTRACT COMMENTS:

• Comprehensive abstract, concise with relevant details and important conclusions about supporting the development of WM in children.

INTRODUCTION COMMENTS:

• Very clear introduction, framing the topic well, and highlighting the novelty/need for the current meta-analysis, good job!

• Minor point – page 6, from line 125 – repeated use of the word different makes sentence less fluent, advise revision.

• Good summary of aims to examine the components of working memory in line with children’s SES.

Response: We appreciate that the repeated use of the word ‘different’ had reduced fluency in this section. We have now addressed this by amending our language (p.7, line 134-137)

METHODS COMMENTS:

• Good use of the PEO framework

• Participants, exposure and outcomes all clearly described

• Clear study design

• Clear study inclusion criteria.

• Excellent search/selection strategy via PROGRESS

• Clear details about data extraction

• Excellent use of risk of bias tools

• Validity/reliability measures very thorough

RESULTS COMMENTS:

• Useful information about study characteristics, especially which countries

• Authors should be commended for the high number of individual participants

• Excellent summaries of heterogeneity, publication bias and sensitivity analyses

• The figures are excellent and help to illustrate the findings well.

• The summary paragraph at the end of the Harvest Plot section is very useful

• Overall the results section is excellent – clearly reported.

DISCUSSION COMMENTS:

• Excellent summary of findings – clearly interprets the wealth of data provided in the paper

• Good use of subheadings

• Good link between parents’ SES, allostatic load and variance in child working memory development

• Also the discussion about stereotype threat was intriguing

OVERALL RECOMMENDATION: 

• An excellent paper, truly well executed and inspiring. Beyond the minor points above, this paper could be accepted in its current form.

• ACCEPT

Response: Thank you very much for your thorough review that highlights the strengths of our paper. This was a very welcome review and we are very grateful to the Reviewer. 

Reviewer #2

Introduction

The authors cover a substantive amount of relevant literature in the introduction to explain the possible mechanisms through which socioeconomic position may affect complex working memory in children under 18 years of age. The authors further highlight the disagreement and conflicting findings within the literature regarding the association between socioeconomic position and complex working memory; this is not justification enough for undertaking the current systematic review and meta-analysis. It would be good to see the authors explain why this information is clinically relevant and what the potential impact it could have for children currently living in lower socioeconomic positions.

Response: We respectfully disagree with the reviewer that the conflicting findings in the extant literature are not sufficient reason, without regard to other issues, to undertake a systematic review and meta-analysis. There is a contribution to be made to our knowledge (IS lower socioeconomic position associated with WM?) irrespective of the size of the effect or impact on the population. A very small attributable risk can have a large population impact if the risk factor is widespread, which is the case for low socioeconomic position. 

We have, however, added a rationale for why clarifying this association is helpful in understanding inequalities in educational attainment to both the introduction and the discussion. We previously included some discussion of this in the introduction, where we explained how critical working memory is for successful learning and therefore for education (p.4 lines 60-68). This, in turn, means it is important to attempt to resolve the dispute in the literature around whether SEP impacts working memory (p. 4, line 70-74). In order to make this more explicit we have added some additional text in the introduction (p.4, line 67-68; p.5, line 80-86; p.7, lines 143-150) and discussion (p. 26, lines, 566-571 and p.29, line 649-652). 

Methodology

The authors state that their protocol is registered on PROSPERO, however, a quick search using the CRD number cited shows no hits. See screen shot below.

Response: We have now included the full search term for the number, which should have included ‘CRD’ at the start (p.9, line 162)

Why have the authors chosen the PEO framework as opposed to the PICOS framework which would include a comparison group/population? PICOS – population, intervention/exposure, comparison, outcome, study design. Based on the full description provided under the study eligibility criteria section, this review is able to meet the conditions for using the PICOS framework.

Response: Whilst we acknowledge the merits of the PICOS framework for a systematic review of interventions, we feel that the PEO framework is a more appropriate framework for our review, as it fully captures all of our concepts of interest. If we had applied the PICOS framework, we would have repeated the criterion of ‘socioeconomic position’ for both exposure and comparison. We also made efforts to describe our eligible study designs following description of the PEO framework (p.9, line 177-183). We also note that this comment contrasts with Reviewer #1’s opinion, as their view was that the PEO framework had been used appropriately. 

Although the PRISMA checklist does not specifically state that study selection needs to be done by two independent reviewers, the PRISMA statement website makes reference to the Cochrane Handbook under the protocol guidelines section. In the Cochrane handbook is explicitly states that all studies from the search should be reviewed by at least two people working independently to check if studies meet the inclusion criteria. 

See https://training.cochrane.org/handbook/archive/v6.1/chapter-04#section-4-6-4

For this study, the second reviewer only reviewed a subset of the excluded studies (not the included studies) as well as the data extraction. In light of this, how do the authors reasonably classify this study selection as being part of a standardized systematic review process?

Response: We accept that it is ideal for all studies in a review to be screened by both reviewers, however, this is often not possible and is not an essential criteria according to Cochrane. The latest Cochrane handbook states that duplicate title and abstract screening is ideal, but that “it is acceptable that this initial screening of titles and abstracts is undertaken by only one person”. Our study fulfils these criteria as a subset of the excluded studies were reviewed, and therefore goes beyond screening by just one person. 

See here: https://training.cochrane.org/handbook/current/chapter-04#section-4-6

Further, whilst we accept that the Cochrane handbook describes best practice for ideal reviewing processes – not all systematic reviews can feasibly follow all of the Cochrane guidelines, and are therefore not considered to be Cochrane reviews. However, such studies are still considered systematic reviews. 

See here: https://www.cochranelibrary.com/about/about-cochrane-reviews

A meta-analysis is typically a metric defined by two relationships, however, from the description of how they conducted the meta-analysis, it’s not explicitly clear that they had two groups to compare in this meta-analysis. Did the authors compare lower SEP to higher SEP in the meta-analysis? If so, I think it would help to explicitly state so. If not, they that also needs to be made clear and also provide a justification for taking the meta-analysis down to only one metric.

Response: We did compare lower SEP to higher SEP groups in the meta-analysis, and have amended the manuscript to make this clearer (p.12, line 255-257). This has also been made clearer by the below suggestion about the meta-analysis figures (p.17, line 367-368, p.18, line 380-381).

In the PRISMA flow diagram it shows that 8192 studies were obtained after duplicates were removed. From this a further 7881 studies were excluded, which would leave you with a total of 311 eligible studies and not 396 as indicated on the flow diagram.

Response: Thank you for picking up on this error in our PRISMA diagram. This error occurred when we updated our search to include studies from a further two years, as we made a mistake when updating the PRISMA diagram to include the numbers from the new search. We have now amended the diagram to show that 8194 studies were screened, and 7798 of these were excluded at the title and abstract stage. The number of full text articles assessed for eligibility was 396. The new diagram has been uploaded with this resubmission.

Results

The result section is presented well. Figures 3 & 4 contain forest plots for which the scale -1 – 3 is provided. It make it simpler for the reader, please can the authors indicate which side of the midline favours the relationship in question.

Response: We have amended the note beneath the figure to explain that the right hand side of the midline favours the higher socioeconomic groups (p.17, line 367-368, p.18, line 380-381).

 Discussion

The discussion merely presents a summary of the findings and doesn’t really create an argument or provide an explanations about the statistical findings.

Response: We respectfully disagree that explanations about statistical findings are not provided in our discussion, and note that Reviewer 1’s opinion on this was that we “clearly interpret the wealth of data” in our discussion. 

Potential explanations of our statistical findings are explored in the initial section of the discussion (see p.23, line 501-504; p.24 line 513 to 516 and line 521-525; p.25, line 532-537 and line 542 to 546). As noted by Reviewer 1, the ‘Directions for future research’ section also highlights possible explanations for the findings by considering mediating factors (p.27, line 596-604), and stereotype threat (p. 26, line 572-588). 

Note also, that in response to the Reviewer 2’s earlier point, we have added to the discussion of the impact of our findings for children’s educational attainment, creating a stronger inference for the importance of our findings (p. 26, lines, 566-571 and p.29, line 650-652).

In the recommendations for future studies the authors state that “it has previously been found that socioeconomic disadvantage is associated with worse working memory in ethnic minority children, whilst ethnic majority children at different levels of socioeconomic risks have similar working memory ability”. Are the authors suggesting that ethnicity itself, inherently affects working memory despite socioeconomic position? Further more they state that the “disadvantage faced by ethnic minority groups may therefore exacerbate the negative association between socioeconomic position and working memory”; can the authors please elaborate on what these disadvantages are?

Response: We are not suggesting anything about causal pathways in making this statement, simply pointing to the associations that are present in the extant literature. Our review highlights that there is not yet sufficient evidence to enable any clarity about whether or not variation in working memory among different ethnic groups cannot be explained entirely by socioeconomic position. The study we cite finds a difference in social gradients for working memory across ethnic groups, and we have clarified this in the revised manuscript (p.26, line 578 to 588). 

In the final sentence of the conclusion the authors reveal the impact of the findings of this review. The potential for this information to be clinically and practically useful should be mentioned much earlier in the manuscript (i.e.: as part of the justification for doing this in the introduction).

Response: Thank you for highlighting this. As stated in response to the Reviewer’s earlier comment, we have revised both our introduction (p.4, line 67-68; p.5, line 80-86; p.7, lines 143-150) and discussion (p. 26, lines, 566-571 and p.29, line 650-652) to emphasise this.

---

## [Decision Letter · Decision Letter 1]

17 Nov 2021

The association between socioeconomic disadvantage and children’s working memory abilities: A systematic review and meta-analysis

PONE-D-21-23197R1

Dear,

We’re pleased to inform you that your manuscript has been judged scientifically suitable for publication and will be formally accepted for publication once it meets all outstanding technical requirements.

Kind regards,

Muhammad Shahzad Aslam, Ph.D.,M.Phil., Pharm-D

Academic Editor

PLOS ONE

Additional Editor Comments (optional):

Reviewers' comments:

Reviewer's Responses to Questions

**Comments to the Author**

1. If the authors have adequately addressed your comments raised in a previous round of review and you feel that this manuscript is now acceptable for publication, you may indicate that here to bypass the “Comments to the Author” section, enter your conflict of interest statement in the “Confidential to Editor” section, and submit your "Accept" recommendation.

Reviewer #1: All comments have been addressed

Reviewer #3: All comments have been addressed

2. Is the manuscript technically sound, and do the data support the conclusions?

Reviewer #1: Yes

Reviewer #3: Yes

3. Has the statistical analysis been performed appropriately and rigorously? 

Reviewer #1: Yes

Reviewer #3: Yes

4. Have the authors made all data underlying the findings in their manuscript fully available?

Reviewer #1: Yes

Reviewer #3: Yes

5. Is the manuscript presented in an intelligible fashion and written in standard English?

Reviewer #1: Yes

Reviewer #3: Yes

6. Review Comments to the Author

Reviewer #1: I am happy with the reviewer's response - it is a well-written paper that deserves publication - good job!

Reviewer #3: I agree with the first reviewer, as I believe the article is very well written, relevant and has a very suitable methodology. The results are well presented and well discussed.

The authors adequately responded to all review suggestions, making this version more complete.

I consider the article to be an asset to the state of the art.

7. PLOS authors have the option to publish the peer review history of their article (what does this mean?). If published, this will include your full peer review and any attached files.

Reviewer #1: **Yes: **Dr Samantha J Brooks

Reviewer #3: **Yes: **Sónia Quintão

---

## [Editor Report · Acceptance letter]

23 Nov 2021

PONE-D-21-23197R1 

The association between socioeconomic disadvantage and children’s working memory abilities: A systematic review and meta-analysis 

Dear Dr. Mooney:

I'm pleased to inform you that your manuscript has been deemed suitable for publication in PLOS ONE. Congratulations! Your manuscript is now with our production department. 

Kind regards, 

on behalf of

Dr. Muhammad Shahzad Aslam 

Academic Editor

PLOS ONE